# Effects of Bilateral Extracephalic Transcranial Direct Current Stimulation on Lower Limb Kinetics in Countermovement Jumps

**DOI:** 10.3390/ijerph20032241

**Published:** 2023-01-27

**Authors:** Zhu Zhiqiang, Wang Wei, Tang Yunqi, Liu Yu

**Affiliations:** 1School of Kinesiology, Shenzhen University, Shenzhen 518000, China; 2School of Kinesiology, Shanghai University of Sport, Shanghai 200438, China; 3College of Art & Design, Shanxi University of Science & Technology, Xi’an 710021, China

**Keywords:** Bi-tDCS, M1, lower limbs, CMJ

## Abstract

Objective: Transcranial direct current stimulation (tDCS) is an effective method for improving sports/exercise performance in humans. However, studies examining the effects of tDCS on jumping performance have reported inconsistent findings, and there is a paucity of studies investigating the effects of tDCS on lower limb energy and kinetics in countermovement jumps (CMJs). Thus, we investigated the effects of tDCS on countermovement jump (CMJ) performance and analysed kinetic variations in the ankle, knee, and hip joints. Methods: In total, 15 healthy young participants randomly received anodal or sham bilateral stimulation of the primary motor cortex (M1). The bilateral tDCS (Bi-tDCS) montage used an intensity of 2 mA for a 20 min monophasic continuous current. Jump height, energy, and lower limb kinetic data in CMJs were collected at pre-stimulation (Pre), post-0 min (Post-0), and post-30 min (Post-30) using a motion capture system and two 3D force plates. Jump height, lower extremity energy, and kinetic variables in CMJs were analysed with two-way repeated-measures ANOVA. Results: (1) Compared to the baseline and sham conditions, the jump height increased except that at Post-30 relative to the sham condition, and the total net energy of lower limbs increased at Post-30 relative to the baseline. (2) Compared to the baseline, the ankle positive energy and net energy decreased in the sham condition; Compared to the baseline and values at Post-0, the maximum ankle torque at Post-30 decreased in both stimulation conditions. (3) The maximum knee power increased compared to the baseline and sham conditions. (4) Regardless of time points, the maximum hip torque in the tDCS condition was higher than it was in the sham condition. Conclusion: Bi-tDCS is an effective method for improving jump height by modulating ankle and knee net energy. The net energy improvement of the lower extremities may be due to variation in the kinetic chain resulting from tDCS-enhanced knee exploration force and maximum hip strength in CMJs. The effects of Bi-tDCS gradually decrease.

## 1. Introduction

Vertical jump performance is a crucial factor in athletic performance [1,2]. Outstanding vertical jump performance enables athletes to perform various high-intensity actions in games [3,4]. Various training approaches have been adopted to improve athletic vertical jump performance, including plyometric, resistance, and vibration training. These approaches have proven useful for enhancing athletic vertical jump performance [2,5]. However, jump performance is related to muscular function and is limited by descending commands from the central nervous system [6]. The primary motor cortex (M1) is engaged extensively in motor control and movement [7,8]. Indeed, M1 excitability regulates sports/exercise performance in humans [9,10,11,12]. Therefore, neuromodulation of M1 may be an effective strategy for enhancing jump performance.

tDCS is a non-invasive neuromodulation technique that involves placing two or more electrodes over the scalp to deliver a constant low-amplitude current flow with a popular parameter range (current density of 0.5–2 mA and stimulation time of 15–20 min) to modulate spontaneous neuronal activity. [13,14]. The effects of tDCS on sports/exercise performance in humans depend on electrode polarity, electrode position, and stimulation intensity [15,16,17]. Anodal/cathodal tDCS will depolarise (anodal) or hyperpolarise (cathodal) the resting membrane potential, which enhances or diminishes the excitability of M1 to increase/decrease its output during exercise. In the study by Leite [18], they investigated the anodal/cathodal tDCS of M1′s effect on the motor task. Their results showed that anodal increased motor task performance, while the cathodal had the opposite effect. Moreover, Angius et al. [19] did not find any effect of bilateral cathodal tDCS on neuromuscular function and performance during cycling. However, Holgado et al. [17] reported that anodal tDCS applied to M1 positively improved individual strength, endurance, motor control, and jumping performance. Halakoo et al. [16] investigated 12 randomised controlled trials on tDCS and compared the effects of unilateral and bilateral tDCS on motor performance. Their analysis revealed that bilateral tDCS was more effective in enhancing performance in healthy individuals. Angius et al. [20] compared the effects of different stimulation montages and reported that extracephalic anodal stimulation over bilateral M1 was more efficient for improving performance compared to cephalic montage. Therefore, bilateral extracephalic montage may be an effective strategy for improving jump performance.

To the best of our knowledge, only five studies to date have investigated the effects of tDCS on jump performance, and these studies had inconsistent results. Three studies demonstrated that anodal tDCS have an ergogenic effect on jump performance [21,22,23]. In the study by Lattari [21], 10 participants completed a randomised, double-blind study of three conditions (anodal, cathodal, and sham). The analysis revealed that jump height, flight time, and muscular peak power were significantly increased after stimulation in the anodal condition. In accordance with these findings, Grosprêtre et al. [22] demonstrated that an extracephalic anodal montage (anodal-M1, cathodal-contralateral shoulder) significantly improved jump performance accompanied by an increase in supraspinal and spinal excitabilities. However, Romero-Arenas et al. [24] and Park et al. [25] did not observe any effect which may be caused by the stimulation montage. Moreover, the neurophysiology study showed that the tDCS effect would last more than 30 min [26]. However, present studies did not investigate the lasting effect of tDCS on jump performance, and little information on the kinetics of the lower limbs during vertical jumping has been provided. Accordingly, we further investigated the impact of tDCS on the biomechanics of jumping. This study aimed to examine the immediate and lasting effects of bilateral extracephalic anodal montage on jump performance and its kinetics pattern. We hypothesised that anodal tDCS would significantly increase jump height and alter the kinetic pattern of the lower extremities, and the effects would last for more than 30 min.

## 2. Methods

### 2.1. Participants

A total of 15 healthy young volunteers (mean ± SD, age: 19.47 ± 1.60 years, height: 182.67 ± 5.63 cm, weight: 69.20 ± 8.22 kg) were recruited for this study. The ideal sample size was calculated using G*Power 3.1.9.2 (Heinrich-Heine-Universität Düsseldorf, North Rhine-Westphalia, Germany) software with a statistical power of 0.8, probability level of 0.05, and effect size of 0.34 [17]. Based on this analysis and a 20% dropout rate, the final sample size included 15 participants.

Participants were recruited through advertisements via the Shanghai University of Sports website, email, and WeChat. The inclusion criteria were as follows: (1) age between 18–30 years old, (2) healthy with normal muscle function, (3) no lower limb injuries within the last 6 months, and (4) sports enthusiasts who spent time doing sports for at least 5 h per week. The exclusion criteria were as follows: (1) previous adverse reaction to transcranial magnetic stimulation (TMS) or tDCS, (2) diagnosis of a neurological disease, and (3) severe head injury. Participants who conformed to the exclusion criteria were excluded. The health of participants was self-reported. Participants who met the basic inclusion criteria visited the biomechanical laboratory twice. Before visiting the laboratory, participants were instructed to refrain from participating in strenuous exercise within 24 h before testing and to avoid consuming any drinks containing stimulants such as alcohol or caffeine within 6 h before testing to reduce the influence of physical activity or beverages on their performance. All participants were informed of the experimental procedures and signed a consent document approved by the Institutional Review Board of the Shanghai University of Sport (102772019RT020).

### 2.2. Experimental Protocol

Each participant visited the laboratory twice to complete the biomechanical assessment of CMJs. At each visit, the participants randomly received anodal or sham stimulations. The two visits were separated by at least 1 week to wash out carry-over effects. Upon arrival at the biomechanical laboratory, participants were required to complete the pre-test preparation, which included cleaning the skin and pasting reflective markers on the lower limbs. Once the preparation was completed, participants warmed up on a treadmill at a speed of 10 km/h for 10 min before testing. After 5 min of rest, researchers collected biomechanical Pre data of the lower extremity in CMJs. Following active or sham stimulation, biomechanical data of the lower extremities in CMJs were collected again at Post-0 and Post-30.

### 2.3. Intervention

The intervention was performed in a quiet room near a biomechanics laboratory. The DC-STIMULATOR MR (NeuroConn, Ilmenau, Germany) was used for tDCS in anodal tDCS or sham sessions. Two sessions operated concurrently during the day. The bi-hemispheric stimulation montage of tDCS was similar to that reported by Anguis et al. [19]. In the active stimulation, two anodal rubber electrodes centre were placed on the bilateral motor cortex C3 and C4, which were determined by a 10–20 positioning cap [27], and two cathodal electrodes were placed on the ipsilateral shoulders. Rubber electrodes (5 × 7 cm in size) were inserted into a saline-soaked sponge pocket before placing them in the target area. The current was ramped up to 2.0 mA over 30 s before 20-min continuous active stimulation and then ramped down to 0 mA over 30 s. In the sham condition, the parameters were the same as those for the anodal stimulation, but only a single 120 s constant active stimulation was applied. During each session, the researcher administering the procedure was aware of the stimulation type based on codes. Both the participants and study workers were blinded to the stimulation condition. The electrical resistance displayed between 5–10 Ω.

### 2.4. Data Collection

#### 2.4.1. Sagittal Kinematics Data

A 3D motion capture system (Vicon T40, Oxford Metrics, Oxford, UK) was used to collect lower-extremity sagittal plane kinematics with 16 infrared cameras at a sampling rate of 240 Hz. In total, 23 retroreflective makers were used to define the foot, leg, and thigh segments [28,29]. The sagittal plane angles of the hip, knee, and ankle joints are presented in Figure 1.

#### 2.4.2. 3D Force Plates

Two 90 × 60 × 10 cm 3D force plates (Kistler 9287 B, Kistler Corporation, Switzerland) were used to collect the ground reaction force at a sample rate of 1200 Hz. Kinetic variables, including (1) maximum and minimum torque; (2) maximum and minimum power; and (3) energy generation, energy absorption, and net energy in CMJs for the hip, knee, and ankle joints, were collected. The force plates and Vicon system data were synchronised by the terminal box of an A/D converter.

### 2.5. Data Analysis

#### 2.5.1. Sagittal Plane Kinematics

The trajectory of the reflective makers was filtered with a fourth-order Butterworth low-pass filter at a cut-off frequency of 7 Hz. Ankle, knee, and hip variables in the sagittal plane were calculated using Visual 3D software (4.00.20, C-Motion Inc., MD, USA). Jump height was calculated using V022g (where $V_{0}$ is the vertical take-off velocity).

#### 2.5.2. Joint Kinetics

Joint variables for the hip, knee, and ankle were calculated using inverse dynamic analysis. Joint torque included maximum and minimum torques ($M_{Max}\;$ and $M_{Min}$). Joint power included maximum and minimum powers ($P_{Max}\;$ and $P_{Min}$). The value was calculated as the product of instantaneous internal joint torque and instantaneous angular velocity:(1)$$P_{j}=M_{j}*\omega_{j}(t)$$
where $M_{j}$ refers to the joint moment, and $\omega_{j}$ is the joint angular velocity.

#### 2.5.3. Joint Energy

Joint energy refers to the joint work amplitude within a duration (time-integrated power) [30] and was calculated as follows:(2)$$Negtive\;energy\left(Positive\;energy\right)=\int_{t1}^{t2}P\left(t\right)\cdot \;dt$$
(3)Net energy=Positive energy+Negtive energy
where the negative power occurring during an eccentric contraction (Squatting to Stretching) is referred to as the negative energy, and the positive power occurring during a concentric contraction (Stretching to Take-off) is referred to as the positive energy. Net energy refers to the net energy consumed during the squat to take-off.

### 2.6. Statistical Analysis

SPSS (version 26.0; SPSS Inc., Chicago, IL, USA) was used for statistical analysis. All data are expressed as mean (SD) and were normally distributed based on the Shapiro–Wilk test. Two-way repeated-measures analysis of variance (ANOVA) was used to test the main and interaction effects (intervention and time) for jump height and kinetic variables in CMJs. If there was a significant interaction, post hoc analyses were performed for further analysis. The significance level was set at ɑ = 0.05.

## 3. Results

A total of 15 participants randomly and repeatedly received either tDCS or sham stimulation. All participants completed the tDCS session, and 13/15 (87%) completed the sham session, 2/15 (13%) dropped out in the sham session. The uncollected data were filled by multiple imputations. To date, no side effects or risk events have been observed.

### 3.1. Jump Height

Two-way repeated-measures ANOVA revealed a significant time × stimulation type interaction effect on jump height (F_(2, 28)_ = 14.81, *p* < 0.001, $\eta_{p}^{2}=0.51$). Within-group analysis revealed significant differences in the tDCS condition (F_(2, 28)_ = 11.25, *p* < 0.001, $\eta_{p}^{2}=0.45$). Compared with the Pre value, Post-0 and Post-30 were significantly increased by 2.51 cm (*p* = 0.01) and 1.76 cm (*p* = 0.08), respectively. Post hoc analysis revealed that at Post-0, jump height was 3.70 cm higher for the tDCS condition than for the sham condition *p* = 0.034) (Table 1 and Figure 2).

### 3.2. Energy of the Lower Extremities

Analysis of the energy of the lower extremities revealed significant differences in total net energy, ankle positive energy, ankle net energy, and knee net energy of the lower extremities (Table 2).

We observed a significant time × stimulation type interaction effect for the total energy of the lower extremities (F_(2, 28)_ = 5.81, *p* = 0.008, $\eta_{p}^{2}=0.29$). A follow-up within-group analysis revealed a significant difference in the tDCS condition (F_(2, 28)_ = 3.53, *p* = 0.043, $\eta_{p}^{2}=0.20$). Lower limb net energy was 5.70 J higher at Post-30 than at Pre (*p* = 0.037) (Figure 3).

We observed a significant time × stimulation-type interaction effect for positive ankle energy (F_(1.43, 20.03)_ = 6.325, *p* = 0.013, $\eta_{p}^{2}=0.311$). Further, within-group analysis revealed significant differences in the sham condition (F_(1.41, 19.71)_ = 6.19, *p* = 0.014, $\eta_{p}^{2}=0.31$), whereby positive ankle energy was 1.75 J lower at Post-0 and 3.19 J lower at Post-30 than at Pre (*p* = 0.049 and *p* = 0.016, respectively) (Figure 4).

The analysis revealed a significant time × stimulation-type interaction effect for ankle net energy (F_(1.32, 18.51)_ =5.60, *p* = 0.019, $\eta_{p}^{2}=0.30$). Further, the within-group analysis revealed significant differences in the sham condition (F_(2, 28)_ = 7.79, *p* = 0.002, $\eta_{p}^{2}=0.358$), whereby the value was 2.19 J lower at Post-0 and 3.56 J lower at Post-30 than at Pre (*p* = 0.037 and *p* = 0.005, respectively) (Figure 5).

We observed a significant main effect of stimulation condition for knee net energy (F_(1, 14)_ = 5.61, *p* = 0.033, $\eta_{p}^{2}=0.29$). Regardless of time, knee net energy was 7.87 J higher for the tDCS condition than for the sham condition (*p* = 0.033) (Figure 6).

### 3.3. Kinetics of Lower Extremities

Analysis of the kinetics of the ankle, knee, and hip muscles revealed significant differences in maximum ankle torque maximum, knee power, and maximum hip torque (Table 3).

We observed a significant main effect of time for ankle maximum torque (F_(2, 28)_ = 6.14, *p* = 0.006, $\eta_{p}^{2}=0.31$). Further analysis revealed that regardless of stimulation type, maximum ankle torque was 1.00 N·m higher at Pre than at Post−30 (*p* = 0.016) (Figure 7).

The analysis revealed a significant time × stimulation–type interaction effect for maximum knee power (F_(2, 28)_ = 4.78, *p* = 0.016, $\eta_{p}^{2}=0.26$). Within-group analysis revealed significant differences in the tDCS condition (F_(2, 28)_ = 5.65, *p* = 0.009, $\eta_{p}^{2}=0.29$). Compared to the Pre value, the Post-0 and Post-30 values were significantly increased by 61.80 W (*p* = 0.014) and 43.50 W (*p* = 0.017), respectively. Post hoc analysis revealed significant differences at Post-0 (*p* = 0.009) and Post-30 (*p* = 0.022). Compared with that in the sham condition, the maximum power in the tDCS condition was 138.37 W higher at Post-0 (*p* = 0.009) and 112.83 W higher at Post-30 (*p* = 0.022) (Figure 8).

We observed a significant main effect of stimulation condition for maximum hip torque (F_(2, 28)_ = 5.33, *p* = 0.037, $\eta_{p}^{2}=0.28$). Regardless of time, maximum hip torque was 4.55 N·m higher for the tDCS condition than for the sham condition (*p* = 0.037) (Figure 9).

## 4. Discussion

The effects of tDCS on the biomechanical patterns of vertical jumping performance remain unclear. In this study, we investigated the effects of tDCS on jump height, energy, and kinetics of the lower extremities in CMJs. As hypothesised, the results revealed that jump height was significantly increased after tDCS stimulation. In addition to an improvement in jump height, a significant variation in kinetic variables was observed for total net energy of the lower extremities, ankle positive energy and net energy, knee net energy, maximum ankle torque, maximum knee power, and maximum hip torque.

### 4.1. Effect of tDCS on Jump Height

Previous studies have investigated the effects of tDCS on jump performance [20,22,23,24,31]. Our results are consistent with those of Lattari et al. [20], Grosprêtre et al. [22], and Chen et al. [23]. The study by Lattari et al. [20] investigated the acute effects of tDCS on CMJ performance and reported that bilateral anodal tDCS increased CMJ height by 3.9 cm. Grosprêtre et al. [22] examined the effect of tDCS on jump height and leg neuromuscular function in healthy young men. Their results demonstrated that the M1 montage (anodal on M1, cathodal on contralateral shoulder) significantly increased maximum vertical and horizontal jump performance and was accompanied by an increase in supraspinal and spinal excitabilities. However, our results are inconsistent with those of Park et al. [25] and Romero-Arenas et al. [24], who did not observe any effect of tDCS on jump height and peak power. The discrepancy in tDCS effects may be due to the stimulation montages. This speculation is supported by the results of Grosprêtre et al. [22] and Anguis et al. [20]. Grosprêtre et al. [22] applied two montages (dlPFC montage: anodal on the left dorsolateral prefrontal cortex, cathodal in the supraorbital area; M1 montage: anodal on M1, cathodal on the contralateral shoulder). The M1 montage significantly improved jump height and neuromuscular excitability, but no effect was observed in the dlPFC montage. In agreement with these results, Anguis et al. [20] compared extracephalic and cephalic montages and observed that an extracephalic montage (anodal on M1 and cathodal on contralateral shoulder) improved the time to task failure (TTF) during leg isometric performance by 17%, but the cephalic montage (anodal on M1 and cathodal on contralateral dorsolateral prefrontal cortex) did not result in improvements in the same TTF task. Therefore, the extracephalic montage was an effective way to improve jump height.

We also investigated the effects of tDCS 30 min post-stimulation. Jump height was significantly higher than that at baseline in the tDCS condition, but no difference was observed between the two conditions 30 min post-stimulation. This can be explained by a decrease in the lasting effects of tDCS. Gan et al. [31] investigated the effects of tDCS on sustained visual search performance. Compared to the sham condition, anodal tDCS induced significantly higher performance in only 10 min. Further, Matsunaga et al. [32] assessed the lasting effects of anodal tDCS on somatosensory-evoked potentials elicited by median nerve electrical stimulation and reported that the amplitude of frontal P22/N30 was significantly increased 10 min after anodal stimulation. Therefore, the effects of tDCS would decrease gradually.

### 4.2. Effects of tDCS on Lower Limb Energy

With regard to energy variables, ankle positive energy and net energy were significantly lower in the sham condition than in the tDCS condition. Further, knee net energy was significantly higher in the tDCS condition than in the sham condition, and total net energy of the lower extremities was significantly higher at Post-30 than at preceding time points. Although there was no significant difference between Pre and Post-0 values in the tDCS condition (*p* = 0.065), the variations in total net energy of the lower extremities were similar to those for jump height. This was partly supported by the tDCS-induced improvement in jump height. Moreover, further analysis of lower limb energy revealed that energy improvements occurred primarily at the ankle and knee.

At the ankle, the energy decrease in the sham condition may have been caused by fatigue attributed to repeated jumps during multiple sessions. However, in the tDCS condition, ankle energy did not decrease, which may be due to the ergogenic effects of tDCS on endurance. Indeed, this assumption is supported by previous literature. Angius et al. [19] demonstrated that an extracephalic montage (anodal over M1, cathedral over contralateral shoulder) significantly improved TTF by 23% in cycling tasks. Moreover, another study by Angius et al. [20] applied the same extracephalic M1 montage and reported that anodal stimulation elicited TTF improvements by 17% during isometric leg exercise. Therefore, the effects of tDCS delay fatigue in the ankle.

We observed that the net energy of the knee was significantly higher in the tDCS condition than in the sham condition. Our results are similar to those reported by Workman et al. [33], who examined the effects of tDCS on the torque and work of the leg muscles during fatigue using isokinetic fatigue testing. Compared to the sham condition, tDCS increased torque and work in knee extension. The authors concluded that the tDCS effects might be due to altered recruitment/discharge rate of the motor unit or cortical excitability. In addition, Ma et al. [34] investigated the effects of tDCS on rowing athlete strength and resting-state brain function. tDCS increased knee isokinetic muscle strength, increased the amplitude of low-frequency fluctuation values in the right precentral gyrus, and increased regional homogeneity values in the left paracentral lobule. These data suggested that tDCS increased M1 excitability to enhance athlete performance. In summary, tDCS improved jumping performance by modulating the net energy of the ankle and knee. At the ankle, tDCS did not significantly improve ankle net energy but effectively prevented the decline of net energy. In contrast, net energy was significantly improved at the knee.

### 4.3. Effects of tDCS on Lower Limb Kinetics

For kinetic variables, maximum knee power was significantly higher at Post-0 and Post-30 than at Pre and in the sham condition. In both conditions, maximum ankle torque was significantly lower at Post-30 than at Pre. Moreover, maximum hip torque was significantly higher in the tDCS group than in the sham group. At the knee, maximum power at Post-0 and Post-30 was significantly improved compared with that at Pre and in the sham condition. Our results suggest that bilateral M1 tDCS increases the knee’s explosive force. This is similar to the findings of Lattari et al. [20] and Lu et al. [35]. Lattari and colleagues investigated the effects of anodal tDCS on lower-limb muscle peak power and demonstrated that CMJs increased by 6.8%. Moreover, Lu et al. [35] reported that the force development rate of the knee in extension and flexion was significantly greater than that at baseline in the tDCS condition. In this regard, the improvement in knee power partly supported the increase in knee net energy.

In both conditions, maximum ankle torque was significantly lower at Post-30 than at Post-0, which may be attributed to the fatigue induced by repeated jumps and the decrease in tDCS after-effects. Indeed, previous studies have reported a decrease in tDCS after-effects. Matsunaga et al. [32] investigated the after-effects of tDCS on somatosensory evoked potentials and observed that evoked potentials at frontal P22/N30, P25/N33, and N33/P40 were significantly decreased 10 min after tDCS stimulation. Nitsche and Paulus [13] and Liebetanz et al. [36] investigated the after-effects of tDCS using TMS and reported that the after-effects of tDCS for improving cortex excitability decreased with time.

Maximum hip torque was significantly higher in the tDCS group than in the sham group. The enhancement of hip torque may be due to the effects of tDCS on the lower limb chain. In our study, the anodal electrode (5 cm × 7 cm) was placed over M1. This approach enables stimulation of a large region of the motor cortex, which may affect the performance of the entire lower limb rather than just a single joint [19]. Further, it may improve jump performance via complex coordination of the entire kinetic chain [37,38,39]. Park et al. [25] reported that tDCS significantly improved the spike performance of professional volleyball players and suggested that tDCS can enhance the motor coordination performance of professional athletes.

### 4.4. Mechanism

The mechanism by which tDCS improves performance remains unclear. Previous studies have suggested that tDCS may decrease the resting membrane electric potential and concentration of γ-aminobutyric acid (GABA) to improve M1 excitability and enhance individual performance [12,14,40,41,42,43,44]. The different effects between tDCS montages may arise from the direction of the current flow. Studies by Rawji et al. [43] and Hannh et al. [44] provide evidence in support of this. Rawji et al. [43] determined the specific effect of the current flow direction in the brain during tDCS and reported that cortical excitability was affected by the relative current direction to the column and axon pathways. Moreover, Hannh et al. [44] examined the directional effects of tDCS on cortical excitability and motor behaviours. Their results revealed that the orientation of current flow through a cortical target impacted both neurophysiological and behavioural outcomes. Therefore, the current flow direction of tDCS may be a key factor in the effect of tDCS, and the stimulus of the tDCS electric field is necessary before intervention.

In this study, 4 of the 15 participants (26.7%) did not exhibit a significant effect of tDCS on jump height, which may be due to inter-individual variability. Laakso et al. [45] examined the effects of individual anatomical features on the efficiency of tDCS and observed that cerebrospinal fluid thickness was a key influencing factor. In this regard, a thicker cerebrospinal fluid layer significantly decreased electric field strength. Therefore, individualised targeted methods can be used to improve the effects of tDCS, such as applying functional magnetic resonance imaging (fMRI) to determine an individual’s anatomical features and using finite element methods to identify target areas of the electric field.

### 4.5. Limitations

Our study has several limitations. First, the Bi-tDCS montage was unable to provide simulation data for tDCS electrical current propagation. Further studies should simulate the exact electrical field intensity of the target area to optimise the effects of tDCS. In addition, the 5 cm × 7 cm electrodes used in the present Bi-tDCS montage were too large to specifically isolate the target area. Evidence suggests that multiple smaller electrode stimulation montages may be a better way to improve current focus on the target area [46]. In addition, the present study lacked neurophysiological evidence to explain the effectiveness of tDCS. Harnessing additional techniques such as fMRI and TMS to investigate neurophysiological changes will provide greater insight into the neurobehavioural effects of tDCS.

## 5. Conclusions

Our results suggest that Bi-tDCS is an effective method for improving jump height by modulating the net energy of the ankles and knees. The net energy improvement in the lower extremities may be due to variations in the kinetic chain resulting from tDCS-enhanced knee exploration force and maximum hip strength in CMJs. In this regard, the effectiveness of Bi-tDCS decreases gradually. Future studies applying simulation data, multiple electrode montages, and neurophysiological and decomposition EMG techniques may improve the efficacy of tDCS.

## Figures and Tables

**Figure 1 ijerph-20-02241-f001:**
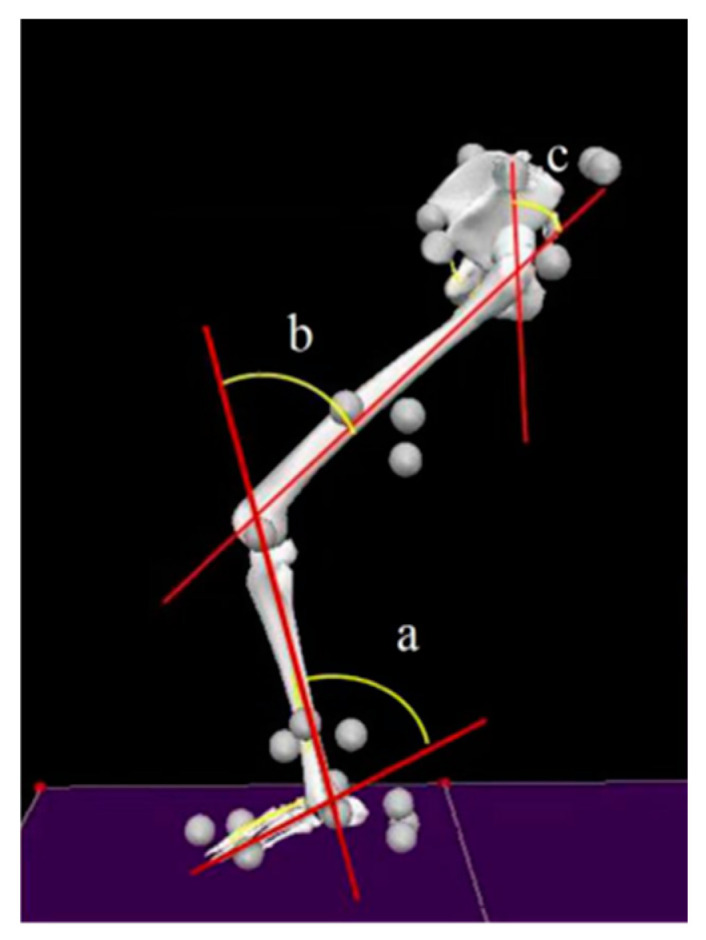
Placements and joint angles of the pelvis, thigh, shank, and foot: (a) angle of the ankle, (b) angle of the knee, (c) angle of the hip.

**Figure 2 ijerph-20-02241-f002:**
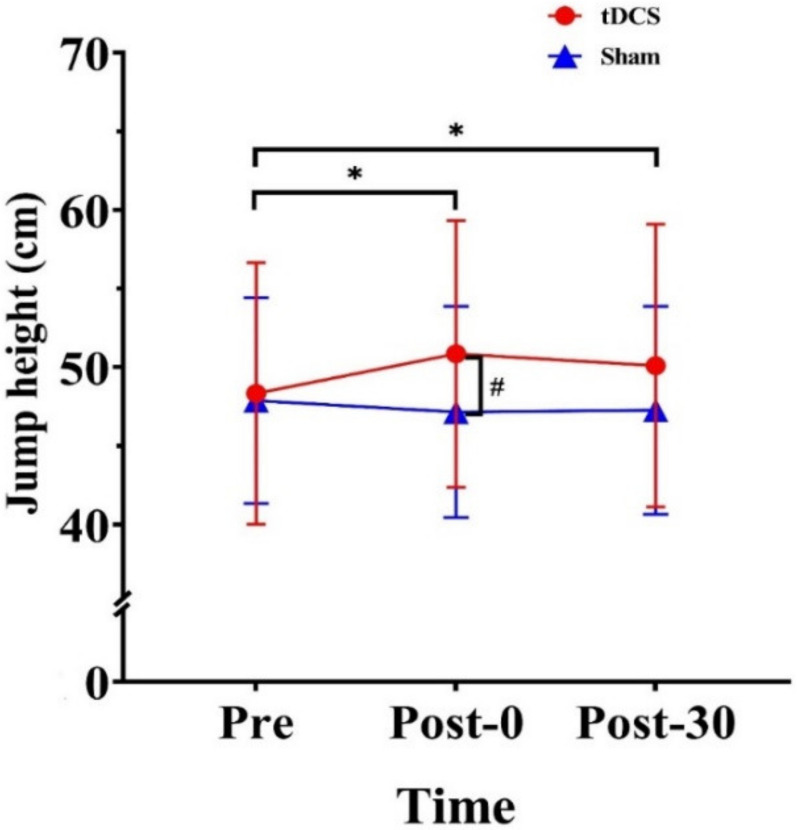
Effects of tDCS on jump height in CMJs. * indicates a significant difference between times. # indicates a significant difference between conditions. Pre: pre-stimulation, Post-0: post-0 min, Post-30: post-30 min.

**Figure 3 ijerph-20-02241-f003:**
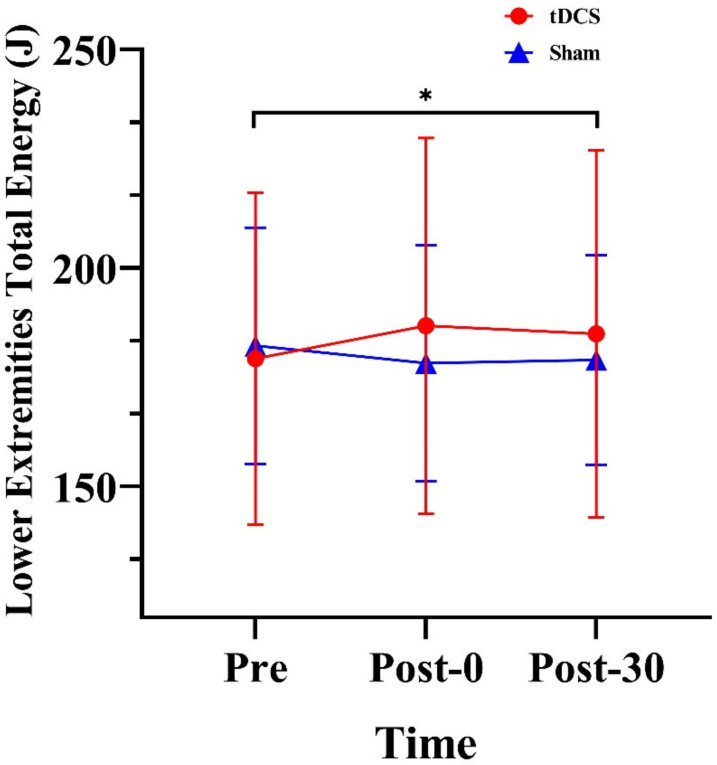
Effects of tDCS on lower limb net energy in CMJs. * indicates a significant difference between times, Pre: pre-stimulation, Post-0: post-0 min, Post-30: post-30 min.

**Figure 4 ijerph-20-02241-f004:**
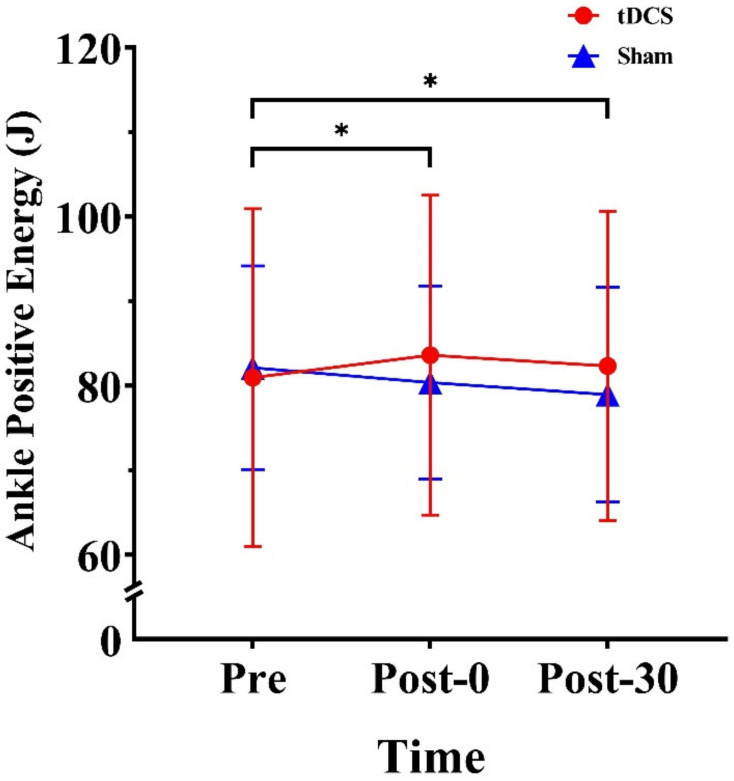
Effects of tDCS on ankle positive energy in CMJs. * indicates a significant difference between times. Pre: pre-stimulation, Post-0: post-0 min, Post-30: post-30 min.

**Figure 5 ijerph-20-02241-f005:**
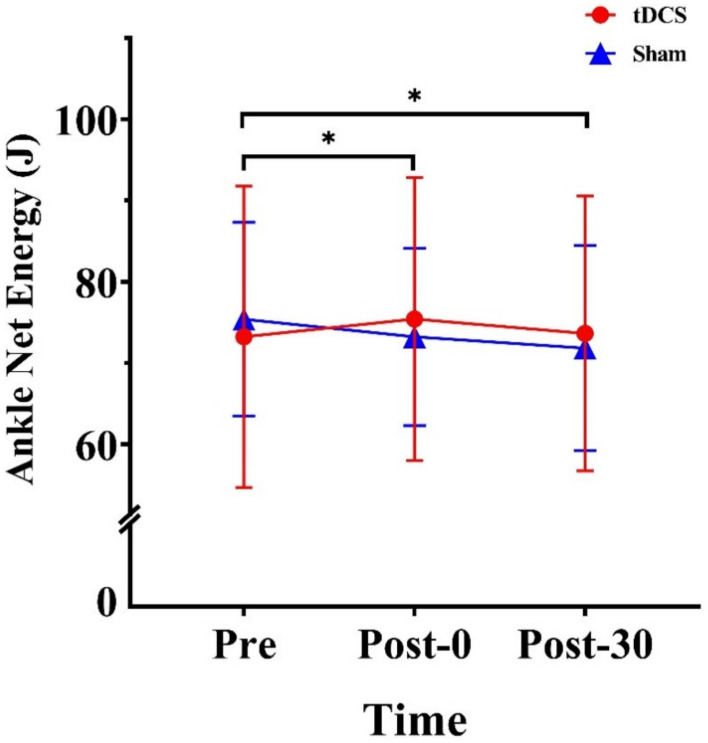
Effects of tDCS on ankle net energy in CMJs. * indicates a significant difference between times. Pre: pre-stimulation, Post-0: post-0 min, Post-30:post-30 min.

**Figure 6 ijerph-20-02241-f006:**
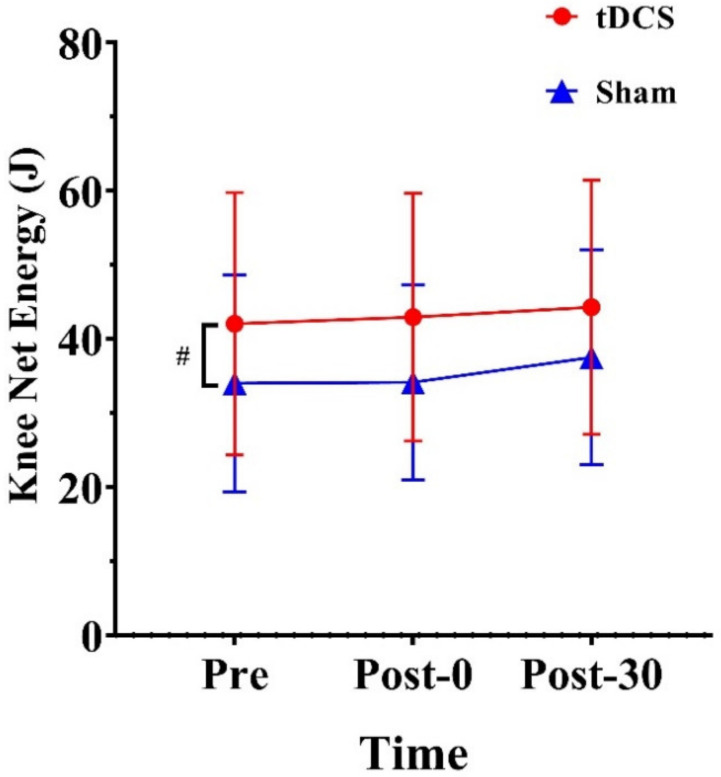
Effects of tDCS on knee net energy in CMJs. # indicates a significant main effect of condition. Pre: pre-stimulation, Post-0: post-0 min, Post-30: post-30 min.

**Figure 7 ijerph-20-02241-f007:**
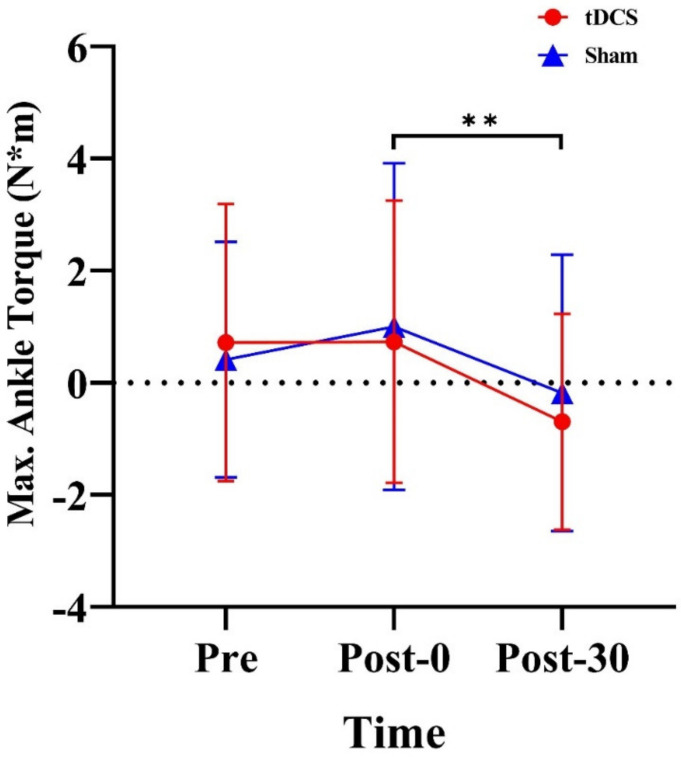
Effects of tDCS on maximum ankle torque in CMJs. ** indicates a significant main effect of time. Max., maximum. Pre: pre-stimulation, Post-0: post-0 min, Post-30: post-30 min.

**Figure 8 ijerph-20-02241-f008:**
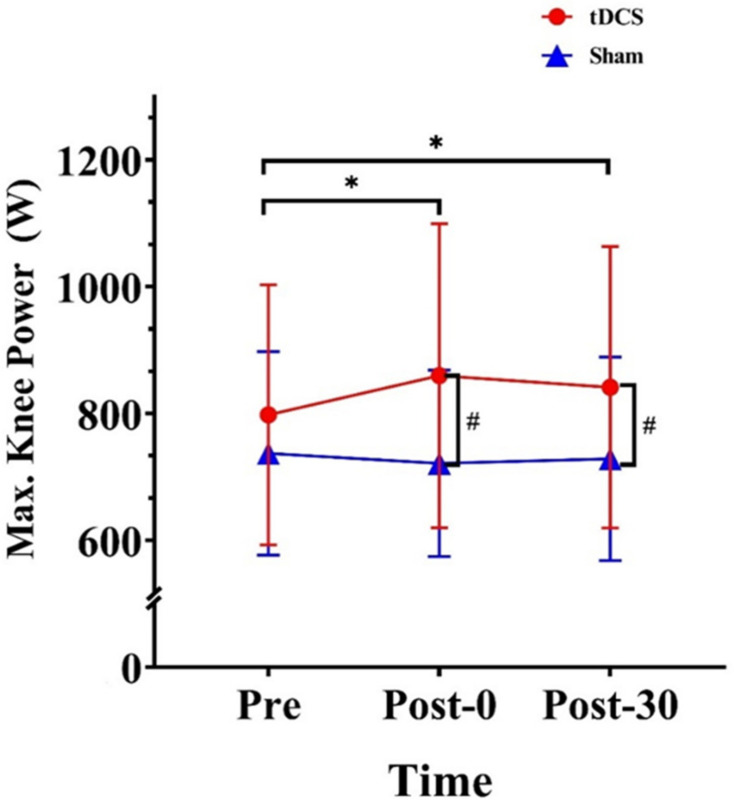
Effects of tDCS on maximum knee power in CMJs. * indicates a significant difference between times. # indicates a significant difference between conditions. Max., maximum. Pre: pre-stimulation, Post-0: post-0 min, Post-30: post-30 min.

**Figure 9 ijerph-20-02241-f009:**
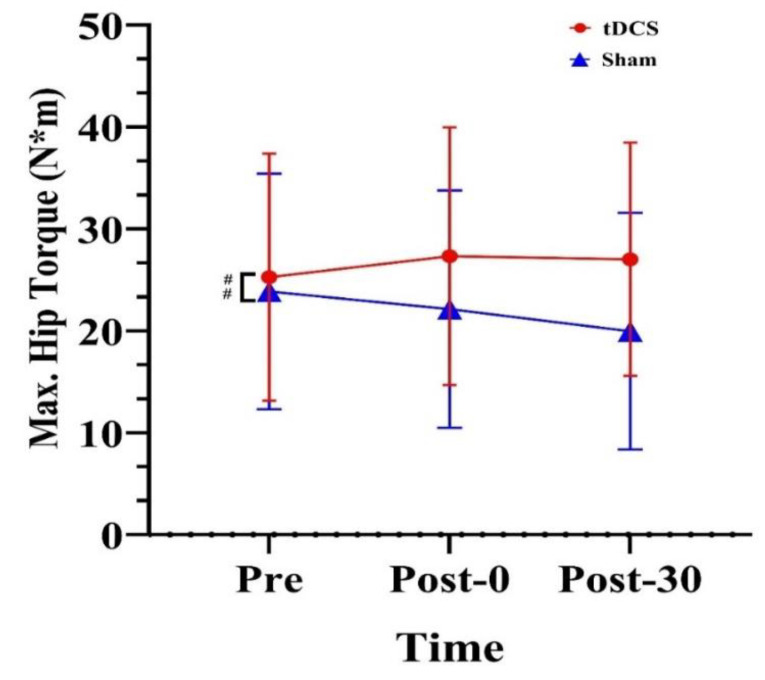
Effects of tDCS on maximum hip torque in CMJs. ## indicates a significant main effect of condition. Max.: maximum, Pre: pre-stimulation, Post-0: post-0 min, Post-30: post-30 min.

**Table 1 ijerph-20-02241-t001:** Jump height during CMJs (*N* = 15, Mean (SD)).

Variables	tDCS			Sham		
	Pre	Post-0	Post-30	Pre	Post-0	Post-30
Jump height (cm)	48.34 (8.32)	50.85 (8.48)	50.10 (8.99)	47.89 (6.54)	47.16 (6.71)	47.26 (6.61)

**Note:** Pre: pre-stimulation, Post-0: post-0 min, Post-30: post-30 min.

**Table 2 ijerph-20-02241-t002:** Energy of the lower extremities in CMJs (*N* = 15, Mean (SD)).

Joints	Variables	tDCS			Sham		
		Pre	Post-0	Post-30	Pre	Post-0	Post-30
Ankle	Negative energy (J)	−7.72 (3.57)	−8.2 (4.00)	−8.7 (3.92)	−6.73 (2.56)	−7.16 (2.60)	−7.10 (2.70)
	Positive energy (J)	80.96 (19.97)	83.61 (18.92)	82.34 (18.30)	82.13 (12.04)	80.38 (11.42)	78.95 (12.69)
	Net energy (J)	73.24 (18.55)	75.41 (17.43)	73.64 (16.91)	75.41 (11.93)	73.22 (10.92)	71.84 (12.65)
Knee	Negative energy (J)	−57.16 (19.22)	−55.85 (21.26)	−56.47 (21.27)	−55.25 (19.33)	−55.86 (20.26)	−54.79 (20.33)
	Positive energy (J)	97.21 (25.06)	98.79 (23.69)	100.76 (23.51)	89.27 (16.74)	90.01 (19.95)	92.32 (18.74)
	Net energy (J)	42.05 (17.71)	42.88 (16.61)	44.29 (17.171)	34.01 (14.65)	34.15 (13.15)	37.52 (14.52)
Hip	Negative energy (J)	−57.60 (12.65)	−55.94 (16.55)	−58.50 (19.84)	−53.86 (13.49)	−52.51 (15.48)	−53.58 (15.44)
	Positive energy (J)	121.55 (29.64)	124.34 (36.75)	125.51 (40.21)	126.63 (33.46)	123.33 (31.48)	123.16 (30.88)
	Net energy (J)	63.95 (22.26)	68.36 (25.83)	67.01 (24.98)	72.77 (21.36)	70.82 (18.43)	69.58 (18.67)
Total net energy of lower limbs (J)		179.23 (38.55)	186.75 (43.25)	184.93 (42.57)	182.19 (27.13)	178.19 (27.84)	178.95 (24.40)

**Note:** Pre: pre-stimulation, Post-0:post-0 min, Post-30:post-30 min, CMJs, countermovement jumps, SD: standard deviation.

**Table 3 ijerph-20-02241-t003:** Kinetics of lower extremities in CMJs (*N* = 15, Mean (SD)).

Joints	Variables	tDCS			Sham		
		Pre	Post-0	Post-30	Pre	Post-0	Post-30
Ankle	Max. torque (N·m)	0.72 (2.47)	0.73 (2.52)	−0.69 (1.93)	0.41 (2.10)	1.0 (2.91)	−0.18 (2.47)
	Min. torque (N·m)	−142.43 (50.03)	−144.10 (47.54)	−146.12 (53.17)	−145.84 (39.19)	−137.91 (34.37)	−137.2 (34.53)
	Max. power (W)	1052.39 (272.87)	1043.69 (257.60)	1032.09 (259.61)	1103.71 (195.94)	1054.74 (145.32)	1035.89 (167.78)
	Min. power (W)	−47.73 (26.58)	−51.63 (28.17)	−50.94 (24.68)	−41.36 (19.43)	−43.04 (15.54)	−38.28 (14.24)
Knee	Max. torque (N·m)	128.18 (31.37)	126.58 (30.66)	128.17 (27.69)	115.29 (18.38)	119.95 (28.98)	121.99 (26.64)
	Min. torque (N·m)	−47.67 (19.44)	−47.98 (18.84)	−48.23 (17.13)	−50.69 (18.50)	−47.00 (14.04)	−45.21 (17.19)
	Max. power (W)	798.03 (205.03)	859.82 (239.87)	841.52 (222.00)	737.42 (160.51)	721.45 (146.92)	728.69 (160.33)
	Min. power (W)	−650.24 (274.87)	−662.61 (269.32)	−663.60 (252.46)	−696.23 (244.16)	−654.87 (185.87)	−627.68 (230.95)
Hip	Max. torque (N·m)	25.27 (12.13)	27.32 (12.64)	27.03 (11.44)	23.86 (11.56)	22.13 (11.63)	19.97 (11.61)
	Min. torque (N·m)	−163.46 (33.78)	−165.56 (31.27)	−166.31 (31.52)	−158.28 (22.14)	−157.93 (22.60)	−155.82 (21.96)
	Max. power (W)	578.89 (135.86)	604.53 (154.24)	597.44 (174.26)	588.94 (125.72)	585.02 (117.72)	576.28 (117.54)

Note: Max.: maximum, Min., minimum.

## Data Availability

All experimental data, together with relevant analysis scripts and files, are available upon request from the author (e-mail: zhuzhiqiang@szu.edu.cn).

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
