# Peer review of "Effects of Bilateral Extracephalic Transcranial Direct Current Stimulation on Lower Limb Kinetics in Countermovement Jumps"

_ijerph, 2023, doi:10.3390/ijerph20032241_

Round 1
Reviewer 1 Report
In the inclusion criteria, you say “healthy young”, i.e. without concomitant diseases. “No lower limb injuries within the last six months” – limb injuries are within the exclusion criteria. In the exclusion criteria, you write about neurological diseases and severe head injury. These are no longer healthy volunteers anymore. Especially after a severe head injury. Volunteers after mild, moderate and severe head injury should be excluded from the study.
The abbreviations used should be described under each table and graph.
12th Reference is placed on line 366, whereas it should be after 11th Reference.
Item 24 - not included in the text. There is only the name of the author (Smith et al.)
Author Response
Response to Reviewer 1 Comments
We would like to sincerely thank the reviewers for their helpful recommendations. We have seriously considered all the comments and carefully revised the manuscript accordingly. Revisions are highlighted in the manuscript using a red font to indicate where changes have taken place. We feel that the quality of the manuscript has been significantly improved with these modifications and improvements based on the reviewers’ suggestions and comments. We hope our revision will lead to an acceptance of our manuscript for publication in the International Journal of Environmental Research and Public Health.
- In the inclusion criteria, you say “healthy young”, i.e. without concomitant diseases. “No lower limb injuries within the last six months” – limb injuries are within the exclusion criteria. In the exclusion criteria, you write about neurological diseases and severe head injury. These are no longer healthy volunteers anymore. Especially after a severe head injury. Volunteers after mild, moderate and severe head injury should be excluded from the study.
Respond to 1:
Thank you for your comments. The participants who conformed to the exclusion criteria were excluded from the study. To make it clear, we added a sentence in lines 99-100.
“Participants who conformed to the exclusion criteria were excluded. The health of participants was self-reported.”
- The abbreviations used should be described under each table and graph.
Respond to 2:
Thank you for your suggestion. As you suggested, we described the abbreviations under each table and graph.
- 12thReference is placed on line 366, whereas it should be after 11th
Respond to 3:
Thank you for your suggestion. As you suggested, we added the reference 12th after Reference 11th. (Line 44)
“Indeed, M1 excitability regulates sports/exercise performance in humans [9-12].”
- Item 24 - not included in the text. There is only the name of the author (Smith et al.)
Respond to 4:
Thank you for your reminder. As you reminded us, we checked item 24 and revised the “(Smith et al.)” into “[30]”. (Line 171)
Reference
[30] Smith, G.; Lake, M.; Lees, A.; Worsfold, P. Measurement procedures affect the interpretation of metatarsophalangeal joint function during accelerated sprinting. Journal of sports sciences 2012, 30, 1521-1527, doi:10.1080/02640414.2012.713501.

Reviewer 2 Report
Zhiqiang et al described the effects of extracephalic tDCS on lower limb kinetics in counter movement jumps. The paper was generally well written, but I have a few concerns though.
Major concerns
1. Write the abstract in such a way so as to summarize the important findings without it looking like a gross summary of the results. Please consider taking out the p values, and simply state if there were significant changes when compared to baseline and sham tDCS.
2. In line 48 to 50, please consider expanding the definition of tDCS. Parameter space has expanded over the last decade, and it is not elegant to keep referring to 0.5-2mA and 15-20 min. Current papers have shown efficacy and safety of tDCS for current intensities of up to 4 mA, and stimulation duration of up to 30 mins.
(a) You could modify the sentence to show that this narrower parameter range is more popular, or
(b) Clearly state a better definition by using an expanded parameter space as stated above.
3. There is an issue with how the aim of the study was stated. It is clear the authors only have five previous studies that investigated the effects of tDCS on jump performance. Three out of five studies were reviewed here (Lines 63-75), and two out of the five studies found some significant improvement in JP. Perhaps the authors should consider writing clearly here up to how long these increases in JP lasted. The very last sentence (the hypothesis) was informed by the review of previous literature. It stands to reason that expecting performance to last up until 30 mins is based on these 2 (out of 5) previous studies. This last paragraph forms the foundation of your study, but it was not well written.
4. The authors should please consider adding a little discussion on what the effects of cathodal tDCS might do differently to CMJ performance in the introduction. This makes a strong point for why the decision is to go for anodal and not cathodal tDCS. This means that a little information should be presented on the mechanisms of action of tDCS. The question therefore is, does stimulation with cathodal bilateral montage improve any variable that wasn’t improved by anodal tDCS?. Are there other mechanistic explanations for the tDCS effects on motor performance?.
Minor concerns
1. Please state whether the health of participants was self-reported, or the health check done by a neurologist or an experimenter using a health checklist. (Methods, lines 87-92).
2. Line 115. NeuroConn. Please consider reviewing the manuscript for spelling errors.
3. Line 117. Please use one citation system throughout the manuscript. Anguis et al 2018 or Angius [18].
4. Line 118. Please consider explaining/adding how the C3, and C4 were determined.
Author Response
Response to Reviewer 2 Comments
We would like to sincerely thank the reviewers for their helpful recommendations. We have seriously considered all the comments and carefully revised the manuscript accordingly. Revisions are highlighted in the manuscript using a red font to indicate where changes have taken place. We feel that the quality of the manuscript has been significantly improved with these modifications and improvements based on the reviewers’ suggestions and comments. We hope our revision will lead to an acceptance of our manuscript for publication in the International Journal of Environmental Research and Public Health.
Major concerns
- Write the abstract in such a way so as to summarize the important findings without it looking like a gross summary of the results. Please consider taking out the p values, and simply state if there were significant changes when compared to baseline and sham tDCS.
Respond to major concern 1:
Thank you very much for your positive comments and all the suggestions that you provided. As you suggested, we simplified the statement of results. The simplified sentence is as follows. (Line 22-29)
“Results: (1)Compared to the baseline and sham conditions, the jump height increased except that at Post-30 relative to the sham condition, and the total net energy of lower limbs increased at Post-30 relative to the baseline. (2) Compared to the baseline, the ankle positive energy and net energy decreased in the sham condition; Compared to the baseline and values at Post-0, the maximum ankle torque at Post-30 decreased in both stimulation conditions. (3) The maximum knee power increased compared to the baseline and sham conditions. (4) Regardless of time points, the maximum hip torque in the tDCS condition was higher than it is in the sham condition.”
- In line 48 to 50, please consider expanding the definition of tDCS. Parameter space has expanded over the last decade, and it is not elegant to keep referring to 0.5-2mA and 15-20 min. Current papers have shown efficacy and safety of tDCS for current intensities of up to 4 mA, and stimulation duration of up to 30 mins.
(a) You could modify the sentence to show that this narrower parameter range is more popular, or
(b) Clearly state a better definition by using an expanded parameter space as stated above.
Respond to major concern 2:
Thank you for your reminder. We modified the definition of tDCS. The modified sentence is as follows. (Line 46-49)
“tDCS is a non-invasive neuromodulation technique that involves placing two or more electrodes over the scalp to deliver a constant low-amplitude current flow with a popular parameter range (current density of 0.5-2 mA and stimulation time of 15–20 min) to modulate spontaneous neuronal activity.”
- There is an issue with how the aim of the study was stated. It is clear the authors only have five previous studies that investigated the effects of tDCS on jump performance. Three out of five studies were reviewed here (Lines 63-75), and two out of the five studies found some significant improvement in JP. Perhaps the authors should consider writing clearly here up to how long these increases in JP lasted. The very last sentence (the hypothesis) was informed by the review of previous literature. It stands to reason that expecting performance to last up until 30 mins is based on these 2 (out of 5) previous studies. This last paragraph forms the foundation of your study, but it was not well written.
Respond to major concern 3:
Thank you for your comments. As you suggested, we revised the last paragraph as follows. (Line 66-84)
“ To the best of our knowledge, only five studies to date have investigated the effects of tDCS on jump performance and had inconsistent results. Three studies demonstrated that anodal tDCS have an ergogenic effect on jump performance [21-23]. In the study by Lattari [21], 10 participants completed a randomised double-blind study of three conditions (anodal, cathodal, and sham). The analysis revealed that jump height, flight time, and muscular peak power were significantly increased after stimulation in the anodal condition. In accordance with these findings, Grosprêtre et al. [22] demonstrated that an extracephalic anodal montage (anodal-M1, cathodal-contralateral shoulder) significantly improved jump performance accompanied by an increase in supraspinal and spinal excitabilities. However, Romero-Arenas et al. [24] and Park et al. [25] did not observe any effect, which may be caused by the stimulation montage. Moreover, the neurophysiology study showed that tDCS effect would be lasting more than 30 minutes [26]. But present studies did not investigate the lasting effect of tDCS on jump performance, and little information on the kinetics of the lower limbs during vertical jumping had been provided. Accordingly, we further investigated the impact of tDCS on the biomechanics of jumping. This study aimed to examine the immediate and lasting effects of bilateral extracephalic anodal montage on jump performance and its kinetics pattern. We hypothesised that anodal tDCS would significantly increase jump height and alter the kinetic pattern of the lower extremities, and the effects would last for more than 30 minutes.”
Reference
[21] Lattari, E.; Campos, C.; Lamego, M.K.; Legey, S.; Neto, G.M.; Rocha, N.B.; Oliveira, A.J.; Carpenter, C.S.; Machado, S. Can Transcranial Direct Current Stimulation Improve Muscle Power in Individuals With Advanced Weight-Training Experience? Journal of strength and conditioning research 2020, 34, 97-103, doi:10.1519/jsc.0000000000001956.
[22] Grosprêtre, S.; Grandperrin, Y.; Nicolier, M.; Gimenez, P.; Vidal, C.; Tio, G.; Haffen, E.; Bennabi, D. Effect of transcranial direct current stimulation on the psychomotor, cognitive, and motor performances of power athletes. Scientific reports 2021, 11, 9731, doi:10.1038/s41598-021-89159-7.
[23] Chen, C.H.; Chen, Y.C.; Jiang, R.S.; Lo, L.Y.; Wang, I.L.; Chiu, C.H. Transcranial Direct Current Stimulation Decreases the Decline of Speed during Repeated Sprinting in Basketball Athletes. International journal of environmental research and public health 2021, 18, doi:10.3390/ijerph18136967.
[24] Romero-Arenas, S.; Calderón-Nadal, G.; Alix-Fages, C.; Jerez-Martínez, A.; Colomer-Poveda, D.; Márquez, G. Transcranial Direct Current Stimulation Does Not Improve Countermovement Jump Performance in Young Healthy Men. Journal of strength and conditioning research 2021, 35, 2918-2921, doi:10.1519/jsc.0000000000003242.
[25] Park, S.B.; Han, D.H.; Hong, J.; Lee, J.W. Transcranial Direct Current Stimulation of Motor Cortex Enhances Spike Performances of Professional Female Volleyball Players. Journal of motor behavior 2022, 1-13, doi:10.1080/00222895.2022.2090489.
[26] Fertonani, A.; Miniussi, C. Transcranial Electrical Stimulation: What We Know and Do Not Know About Mechanisms. The Neuroscientist : a review journal bringing neurobiology, neurology and psychiatry 2017, 23, 109-123, doi:10.1177/1073858416631966.
- The authors should please consider adding a little discussion on what the effects of cathodal tDCS might do differently to CMJ performance in the introduction. This makes a strong point for why the decision is to go for anodal and not cathodal tDCS. This means that a little information should be presented on the mechanisms of action of tDCS. The question therefore is, does stimulation with cathodal bilateral montage improve any variable that wasn’t improved by anodal tDCS? Are there other mechanistic explanations for the tDCS effects on motor performance?
Respond to major concern 4:
Thank you for your questions. As you suggested, we added a discussion on the mechanism of tDCS and the effect of cathodal stimulation on performance. The added discussion part is as follows. (Line 51-57)
“Anodal/cathodal tDCS will depolarize (anodal) or hyperpolarize (cathodal) the resting membrane potential which enhances or diminishes the excitability of M1 to increase/decrease its output during exercise. In the study by Leite [18], they investigated the anodal/cathodal tDCS of M1 effect on the motor task. Their results showed that anodal increased motor task performance, while the cathodal had the opposite effect. Moreover, Angius et al. [19] did not find any effect of bilateral cathodal tDCS on neuromuscular function and performance during cycling. However,”
Reference
[18] Leite, J.; Carvalho, S.; Fregni, F.; Gonçalves Ó, F. Task-specific effects of tDCS-induced cortical excitability changes on cognitive and motor sequence set shifting performance. PloS one 2011, 6, e24140, doi:10.1371/journal.pone.0024140.
[19] Angius, L.; Mauger, A.R.; Hopker, J.; Pascual-Leone, A.; Santarnecchi, E.; Marcora, S.M. Bilateral extracephalic transcranial direct current stimulation improves endurance performance in healthy individuals. Brain stimulation 2018, 11, 108-117, doi:10.1016/j.brs.2017.09.017.
Minor concerns
- Please state whether the health of participants was self-reported, or the health check done by a neurologist or an experimenter using a health checklist. (Methods, lines 87-92).
Respond to minor concern 1:
Thank you for your suggestions. As you suggested, we have added a sentence to state the participant’s situation. (Line 98)
“The health of participants was self-reported”
- Line 115. NeuroConn. Please consider reviewing the manuscript for spelling errors.
Respond to minor concern 2:
Thank you very much for your suggestions. As you suggested, we carefully checked the spelling errors in the manuscript, and changed the spelling of “NeuroCnn” to “Neurocnn”. (Line 120)
- Line 117. Please use one citation system throughout the manuscript. Anguis et al 2018 or Angius [18].
Respond to minor concern 3:
Thank you for your suggestions. As you suggested, we checked the manuscript and revised two citation styles as follows.
1、we changed “(Anguis et al 2018)” to “Anguis et al [19]” (Line 122)
2、we changed “(Smith, et al., 2012)” to “[30]” (Line 169)
Reference
[19] Angius, L.; Mauger, A.R.; Hopker, J.; Pascual-Leone, A.; Santarnecchi, E.; Marcora, S.M. Bilateral extracephalic transcranial direct current stimulation improves endurance performance in healthy individuals. Brain stimulation 2018, 11, 108-117, doi:10.1016/j.brs.2017.09.017.
[30] Smith, G.; Lake, M.; Lees, A.; Worsfold, P. Measurement procedures affect the interpretation of metatarsophalangeal joint function during accelerated sprinting. Journal of sports sciences 2012, 30, 1521-1527, doi:10.1080/02640414.2012.713501.
- Line 118. Please consider explaining/adding how the C3, and C4 were determined.
Respond to minor concern 4:
Thank you for your reminder. The position of C3 and C4 were determined by a 10-20 positioning cap according to the study of Herwig et al. (2003). We modified the sentence as follows.
“Two anodal rubber electrodes center were placed on the bilateral motor cortex C3 and C4 which were determined by a 10-20 positioning cap [27]. ” (Line 125 )
Reference
[27] Herwig, U.; Satrapi, P.; Schönfeldt-Lecuona, C. Using the international 10-20 EEG system for positioning of transcranial magnetic stimulation. Brain topography 2003, 16, 95-99, doi:10.1023/b:brat.0000006333.93597.9d

Round 2
Reviewer 2 Report
I really appreciate the effort taken by the authors to revise the manuscript.
The manuscript looks very good now. I however think there are a few more things missing that must be addressed.
1. Line: 123: I would be happy if the issue of the stimulator brand name is settled. The stimulator is NeuroConn from Germany. Unless there is a re-branding that I am not aware of. You still wrote Neurocnn. You are missing the 'o'. Is the brand name Neurocnn?. I've always known the DC stimulator from Ilmenau to be NeuroConn. Please this is critical error if the name is NeuroConn. Please consider changing it.
2. Lines 188-190: In the Results section, you stated that only 13/15 (87%) had sham stimulation too. Please consider providing a reason for this. Also, if they were randomized into tDCS and sham, does that mean there were two groups?. Did the sham group also receive tDCS?. This is not clear from the few lines in the beginning of the Result section. Please clarify.
3. There is no need to explain the acronym tDCS everytime under all the figures. You can simply use the tDCS without writing the full meaning each time you use it.
Everything else is fine.
Author Response
Response to Reviewer 2 Comments
We very sincerely thank the reviewer for the helpful recommendations. We have seriously considered all the comments and carefully revised the manuscript accordingly. Revisions are highlighted in the manuscript using a red font to indicate where changes have taken place. With the help of the reviewer, We feel that the quality of the manuscript has been improved again. We hope our revision will lead to an acceptance of our manuscript for publication in the International Journal of Environmental Research and Public Health.
- Line: 123: I would be happy if the issue of the stimulator brand name is settled. The stimulator is NeuroConn from Germany. Unless there is a re-branding that I am not aware of. You still wrote Neurocnn. You are missing the 'o'. Is the brand name Neurocnn?. I've always known the DC stimulator from Ilmenau to be NeuroConn. Please this is critical error if the name is NeuroConn. Please consider changing it.
Respond to 1:
Thanks very much for your reminder, as you reminded us, we checked the stimulator brand name again and found the name of the stimulator is “NeuroConn”. We changed the “Neurocnn” to “NeuroConn”. (Line 122)
- Lines 188-190: In the Results section, you stated that only 13/15 (87%) had sham stimulation too. Please consider providing a reason for this. Also, if they were randomized into tDCS and sham, does that mean there were two groups?. Did the sham group also receive tDCS?. This is not clear from the few lines in the beginning of the Result section. Please clarify.
Respond to 2:
Thank you for your question. Our study is a repeated randomized control design. Each participant was asked to randomly accept active or sham tDCS. Two participants (2/15) in the study dropped out in the sham session. The uncollected data were filled by multiple imputations. To clarify, we modified the sentence as follows. (Line 187-190)
“A total of 15 participants randomly and repeatedly received either tDCS or sham stimulation. All participants completed the tDCS session, and 13/15 (87%) completed the sham session, 2/15 (13%) dropped out in the sham session. The uncollected data were filled by multiple imputations.”
- There is no need to explain the acronym tDCS everytime under all the figures. You can simply use the tDCS without writing the full meaning each time you use it.
Respond to 3:
Thank you for your suggestion. As you suggested, we deleted the full meaning of tDCS in each figure and table.
